# Circulating microRNAs in Hidradenitis Suppurativa

**DOI:** 10.3390/genes13091544

**Published:** 2022-08-26

**Authors:** Bruna De Felice, Concetta Montanino, Marta Mallardo, Graziella Babino, Edi Mattera, Giovanni Ragozzino, Giuseppe Argenziano, Aurora Daniele, Ersilia Nigro

**Affiliations:** 1Department of Environmental, Biological and Pharmaceutical Sciences and Technologies (DISTABIF), University of Campania Luigi Vanvitelli, Via Vivaldi 43, 81100 Caserta, Italy; 2CEINGE-Biotecnologie Avanzate, Via G. Salvatore, 486, 80145 Napoli, Italy; 3Dermatology Unit, Università Degli Studi della Campania “Luigi Vanvitelli”, 80126 Naples, Italy; 4Department of Internal and Experimental Medicine and Surgery Unit of Internal Medicine, Università degli Studi della Campania “Luigi Vanvitelli”, Via Pansini, 80126 Napoli, Italy; 5Dipartimento di Medicina Molecolare e Biotecnologie Mediche, Università degli Studi di Napoli “Federico II”, 80126 Napoli, Italy

**Keywords:** miRNA, Hidradenitis suppurativa, real-time qPCR

## Abstract

Hidradenitis suppurativa (HS) is a pathology characterized by chronic inflammation and skin lesions. The molecular basis of the inflammatory network remains unclear; however, since microRNAs (miRNAs) are involved in the modulation of inflammation, the composition of a micro-transcriptome RNA library using the blood of HS patients was analysed here. The total miRNA expression profiles of miRNAs from HS patients was assayed by real-time qPCR. Here, compared to healthy controls, miR-24-1-5p, miR-146a-5p, miR26a-5p, miR-206, miR338-3p, and miR-338-5p expression was found significantly different in HS. Knowing the significance of the miRNA mechanism in inflammatory and immune progression, we suggest that miRNA profiles found in HS patients can be significant in understanding the pathogenesis modality and establishing efficient biomarkers for HS early diagnosis. In particular, miR-338-5p was closely related to HS invasiveness and production of cytokines and was atypically overexpressed. miR-338-5p may represent a good promise as a non-invasive clinical biomarker for HS.

## 1. Introduction

Hidradenitis suppurativa (HS) is a complex and chronic inflammatory condition of the skin that shows, primarily, a dysregulation of the innate immune system and then an inflammatory state that could cause severe injury, including painful abscesses and draining sinus tracts in numerous regions of the body [1,2]. Factors such as cigarette smoking and obesity are predisposing factors for the development of HS.

Various studies have discovered an important genetic influence; in fact, 30–40% of HS patients reported a family history of HS [3]. In different populations, an autosomal dominant form of HS with has been described 100% penetrance [4]. The HS rate has increased to as high as 11.4 per 100,000 patients in the population [5]. In the general population, HS predominantly affects women with respect to men with a 3:1 ratio [2]. The basis of HS disease is an over-activated immune-cell infiltration that enhances the inflammatory response through the emission of a considerable quantity of pro-inflammatory (IL-1β, TNF α, IL-17, and INF-α) and anti-inflammatory cytokines (IL-10) and chemokines [6,7].

The inflammation can be chronic and, especially in severe cases, could affect different tissues and organs [8,9]. On the other hand, an immunological “priming” in HS also comes from environmental factors such as, smoking- and obesity-related inflammatory mediators [10,11]. In particular, in obesity, the enlarged adipose tissue also determines a pro-inflammatory environment due to the imbalance in the production of adipokines that contribute to the severity and progression of HS disease [12]. MicroRNAs (miRNAs) are small noncoding RNA molecules of 17–22 nucleotides that regulate gene expression in a post-transcriptional site by bonding with messenger RNAs (mRNAs) inducing translational inhibition and/or decay [13]. MicroRNA apparatuses and the expression of microRNAs themselves are undoubtedly involved in most cellular responses, and altered expression has been made known mainly in cancer but also in inflammatory and autoimmune sicknesses such as psoriasis and ulcerative colitis [14]. Although miRNAs arose as important modulators of inflammation, information about their expression in HS is limited. Significant advancement has been accomplished in the exploration of a crosstalk between miRNAs and metabolic disorders, and recently, a number of miRNAs have been documented to be intricate in many diseases, confirming that their concentration is incredibly high in serum and/or plasma and other body fluids [15]. Based on the importance of miRNA machinery in immune and inflammatory processes [16,17], and owing to previous available outcomes showing altered expression profiles of some miRNA expression in inflammatory HS lesions, we intended to investigate the total miRNA expression profiles of miRNAs in blood from HS patients compared to healthy controls. Here, miR-146a-5p, miR-206, miR338-3p, miR-338-5p, miR-Let 7a, miR-24-1-5p, and miR26a-5p expression was found to be significantly different compared to healthy controls.

## 2. Materials and Methods

### 2.1. Recruitment of Patients

A total of 25 patients (19 females, 6 men), aged 27.07 ± 14.76 years, were recruited from the Dermatology Unit of the Università degli Studi della Campania “Luigi Vanvitelli”. 

Subjects were excluded from our study if they met any of the following criteria: age < 18 years, BMI < 17 or >35, major metabolic disorders (Type 2 Diabetes, cardiovascular disorders, and metabolic syndrome), syndromic HS, the concomitant presence of an inflammatory cutaneous or systemic syndromes and the presence of cancer; also excluded were patients receiving any systemic treatment that could interfere with the considered parameters. Disease staging was based on the three-degree Hurley’s scale. The mean body mass index (BMI) of 28.11 ± 6.19 kg/m^2^ classified our patients as overweight, and the smoker rate was 51%. 

Twelve healthy volunteers were recruited (7 men, 5 females), aged 33.25 ± 11.26 years old, who constituted the control group (BMI = 25.71 ± 2.98). All HS patients fulfilled the established HS diagnostic criteria. An informed consent form has been signed by all the subjects. The Ethic Committee of the Università degli Studi della Campania “Luigi Vanvitelli” approved the research (Prot. 12478/20).

### 2.2. RNA Extraction

Peripheral blood samples were obtained from HS patients and healthy controls. For RNA isolation and purification, the Trizol (Invitrogen, #15596-026) method was used for all samples within 1 h of drawing in order to reduce RNA degradation. The RNA isolated with this protocol comes from all white cells (polymorphonuclear leukocytes and mononuclear cells) and includes an optional DNase digestion step. This standardized RNA isolation practice guarantees high-quality, non-degraded RNA. The quality of RNA samples was checked by identification of 18S rRNA and 28S rRNA peaks through the Agilent 2100 Bioanalyzer platform (Agilent Technologies, Santa Clara, CA, USA).

### 2.3. Small RNA Library Preparation and Sequencing

Three small RNA libraries were constructed as previously described [18]. Total RNA (40 μg) was collected in three pools, each comprising RNA from blood leukocytes of recruited patients. RNA pools were separated, based on their length, in 15% Tris/Borate/EDTA urea polyacrylamide electrophoresis gel, and the sncRNAs of about 18–80 nt were extracted, purified, and cloned. Approximately 1020 cloned sncRNA molecules were independently sequenced for each library. Colony PCR was performed using 5′ and 3′ primers, and the clones with PCR products about 110 bp in length were sequenced.

### 2.4. Reverse Transcription-Quantitative Polymerase Chain Reaction (RT-qPCR) 

cDNA was gained by using stem-loop reverse transcriptase (RT) primers or oligo-dT primers. RNU6B was used as internal control for microRNAs. Real-Time qPCR was performed using the following conditions: 94 °C for 4 min tailed by 40 cycles at 94 °C for 1 min, 56 °C for 1 min, and 72 °C for 1 min. Relative expression levels of miR-146a, miR-206, miR-338-3p, miR-338-5p, miR-Let 7a, miR-24, and miR-39-1p were obtained using the 2^−^^ΔΔ^^Ct^ method. Using the mean ± standard deviation, data were obtained. A two-tailed Student’s *t*-test was performed for group comparisons, and *p* < 0.05 was regarded as statistically significant.

### 2.5. MiRNA Target Prediction

The potential target mRNAs for distinct miRNAs were identified by searching them on public and different databases endowed with prediction algorithms, such as TargetScan (http://targetscan.org), PicTar (http://pictar.mdc-berlin.de, accessed on 4 June 2022), miRBase (http://www.mirbase.org, accessed on 4 June 2022), TarBase (http://microrna.gr/tarbase, accessed on 4 June 2022), and Miranda (http://microrna.sanger.ac.uk/sequences, accessed on 4 June 2022).

mRNAs that were found to be significant in both methods were considered potential significant targets. The string database was used to identify the protein–protein interaction (PPI) network and to perform Gene Ontology and functional annotation [19].

## 3. Results

### 3.1. Anthropometric and Laboratory Investigations

The biochemical and anthropometric parameters of HS patients and controls are shown in Table 1. The mean body mass index (BMI) of HS patients was 28.11 ± 6.19 kg/m^2^, qualifying our patients as overweight. The smoker rate amounted to 51%, and we excluded major metabolic disorders, as well as concomitant inflammatory cutaneous or systemic disorders and any systemic treatment that could interfere with the studied parameters.

### 3.2. qRTPCR microRNA Expression

To characterize the small non-coding RNA (sncRNA) signature from the peripheral blood of HS patients, we evaluated a set of 25 HS cases by sncRNA cloning. For each library, nearly 50,000 small non-coding RNA cloned molecules were sequenced, and we obtained a profile of sncRNAs expressed in blood leukocytes. We were able to isolate seven mature microRNA species (Figure 1).

MicroRNA expression results from blood and serum HS samples were validated by RT-qPCR. Further real-time PCR analysis was performed for miRNAs showing a significant expression difference (*p* < 0.05 and 1.5 fold change) between naïve HS patients compared to controls. miRNA data are displayed in Figure 1.

In all samples, the expression level of let-7 was low, and there were no significant differences between HS and healthy controls. The expression level of microRNAs such as miR-146a-5p, miR-206, miR338-3p, miR-24-1-5p, and miR26a-5p was meaningfully lower in HS compared to healthy controls (*p* < 0.05). On the other hand, miR338-5p (*p* < 0.048) was remarkably overexpressed in HS in comparison to controls. 

Il1, Il6, and TNFα expression was notably higher in HS related to healthy controls (*p* < 0.045) with IL6 showing the highest difference between HS and healthy controls compared to IL1 (*p* < 0.05) (Figure 2). On the other hand, COX-2 expression levels were remarkably lower in HS samples in comparison to controls (*p* < 0.05).

### 3.3. Computational Predictions of the Putative miRNAs Target

MicroRNAs are most interesting for potential target genes, which are studied with the principle that miRNAs can bind mRNAs and target them for transcriptional degradation or translational inhibition. We searched putative miRNA target genes on five different web-accessible miRNA target databases: http://mirsystem.cgm.ntu.edu.tw/; http://geneontology.org/; http://pantherdb.org/webservices/go/overrep.jsp; http://www.targetscan.org/vert_80/; and https://mirtarbase.cuhk.edu.cn (accessed on 9 June 2022).

To identify target mRNAs, we used the results obtained by combining five different bioinformatics tools. Specifically, the target genes found from this method have been submitted to the KEGG pathway and Gene Ontology tools, both implemented in the string database (Figure 3 and Figure 4). 

As shown in Figure 3, concerning the Gene Ontology analysis, we obtained genes for the molecular function. Nevertheless, as shown in Figure 3, we obtained a statistically noteworthy enhancement of a group of genes involved in immune system pathways such as cytokine production, TNF, and IL-6 signaling. Molecular network outputs counting miRNAs isolated in our research and target genes involved mainly in the immune response categories are shown in Figure 4.

In particular, Table 2 records the pathways principally linked to inflammation and mediated by the interleukin signaling pathway, chemokine and cytokine signaling, and neuronal compartments in which microRNA and mRNA targets are involved.

## 4. Discussion

Hidradenitis suppurativa (HS) is a chronic inflammatory pathology. It is characterized by painful purulent skin lesions with progressive destruction of the skin structure. The loss of control of several mechanisms predisposes the tissues to multiple pathologies that drive the evolution of the inflammation [16].

miRNAs are involved in many diseases, including immune and skin disorders. They have been extensively studied as important inflammatory response modulators. For example, miRNAs have an extensive range of significant functions in the immune system, playing a role in IL-10-mediated suppression of TNFα, IL-6, and IL-12 [17]. In the context of inflammation, miRNAs regulate both the development and function of immune cells that are crucial for the activation of tissue stromal cells and production of proinflammatory cytokines and chemokines that recruit monocytes and granulocytes [20]. 

According to extensive genetic analysis, only a minority of patients have a monogenetic diagnosis [21]. Consequently, epigenetic phenomena may influence HS. The epigenetic regulation of gene expression may represent a main contributor to the variable predisposition of such diseases, which is supported by growing evidence [22]. Lesioned skin of HS patients, compared to controls, has shown that miRNAs were differentially expressed. This has led to the hypothesis of a microRNA function in the modulation of the inflammatory response in lesioned skin of HS patients [23]. In the pathogenesis of HS, the expression levels of microRNA were found to be dysregulated in inflamed tissue, but with limitations a lack of information on patient ethnicity and the low number of individuals considered possibly accounting for genetic differences [24].

The research on the clinical influence of microRNAs in human HS is at a very early point. In our study, we established microRNAs in blood from HS patients by comparing the expression levels to healthy controls. In particular, miR-146a-5p, miR-206, miR338-3p, miR-24-1-5p, and miR26a-5p were shown to be modulated in HS patients. miR-338-5p was over-expressed and the other miRNAs were down-regulated. 

In particular, miR-146a-5p is increasingly significant as both an innate and adaptive immunity cell function and modulator of differentiation. The inflammamiR, such as miR-146a-5p, targets a multiplicity of molecules that are part of the NF-κB/NLRP3 pathways. The interaction between miR-146a-5p and IL-6 in aging and most common age-related diseases (ARDs) has been underlined. The measure of inflammaging is the most relevant evidence of circulating inflammamiRs together with IL-6. [25]. Furthermore, there is evidence from a study that the suppression of miR-146a in cystic fibrosis macrophages increases IL-6 production, indicating that miR-146a is functional [26]. Preclinical studies have shown that lacking miR-146a-5p expression generates inflammation, contributing to the pathogenesis of vascular complications in diabetes. MicroRNA-146a-5p (miR-146a-5p) is a significant regulator of inflammatory processes related to interleukin 6 (IL-6) and tumor necrosis factor α (TNF-α) levels. The expression of miR-146a-5p is changed in diabetic organs, and miR-146a-5p deficits have been involved in their pathogenesis [27]. In addition, miR-146a-5p is an accepted marker of inflammation and is related to immune and non-immune inflammatory disorders [25], as well as diseases such as Alzheimer’s, CVD, and type 2 diabetes [28,29]. 

Outcomes have revealed that miR-146a-5p is down-regulated in breast cancer cells. IRAK1, i.e., interleukin-1 receptor-associated kinase 1, is a direct target of miR-146a-5p. Higher expression of IRAK1 in breast cancer tissues stimulates the growth, migration, and invasion of cancer cells [30].

miR-146a-5p has been extensively identified as a regulatory element in the immune response. However, miR-146a-5p may also be included in the development of Alzheimer’s disease (AD), since it has been proved that miR-146a-5pa increases Aβ deposition by triggering oxidative stress through activation of MAPK signaling in mice models and SH-SY5Y cells treated with amyloid β (Aβ) [31].

miR-146a-5p expression in AD patients’ plasma is correlated with age and illness severity, and there is also an effect of sex. Further, related targets of miR-146a in AD are the chemokine receptor 4 (*CXCR4*), the Fas-Associated Death Domain (*FADD*; validated targets), and the microtubule-associated protein tau (MAPT) [32].

MiRNAs present in body fluids and circulating miRNAs are very promising as non-invasive clinical biomarkers. However, the connection between miR-146a-5p and HS is complex and is likely influenced by many other variables. 

On the other hand, the miR-338-5p level was detected by RT-qPCR in all HS patients. The results showed that miR-338-5p expression in HS patients was unusually high compared to the control group.

The results of RT-qPCR stated that the levels of IL-1a, IL-6, and COX2 in HS increased significantly compared with the controls.

Even though miR-338-5p has been shown to be related to several human diseases, as well as rheumatoid arthritis [33], human gastric cancer [34], pulmonary fibrosis [35], lung cancer [36], and glioblastoma [37], the relationship between miR-338-5p and HS has not been described yet. However, we speculate that miR-338-5p stimulates the cell proliferation, invasion, and the production of cytokines IL-1a, IL-6, and COX2 in HS. 

## 5. Conclusions

For the first time ever, we established that miR-338-5p is overexpressed in HS. We assume that miR-338-5p may have a central role in HS pathogenesis. MiR-338-5p was detected as being abnormally up-regulated in HS tissues and strictly linked to invasiveness and the production of cytokines.

Additional experiments could be performed to understand the molecular network in Hidradenitis and to determine whether other types of cells may contribute to the development and progression of the disease. miR-338-5p has the potential to be employed for the management and treatment of HS. Since miRNAs are found in body fluids, circulating miRNAs and miR-338-5p have good potential as clinical non-invasive biomarkers, even in HS. 

Supplementary research is needed to settle such assumptions.

## Figures and Tables

**Figure 1 genes-13-01544-f001:**
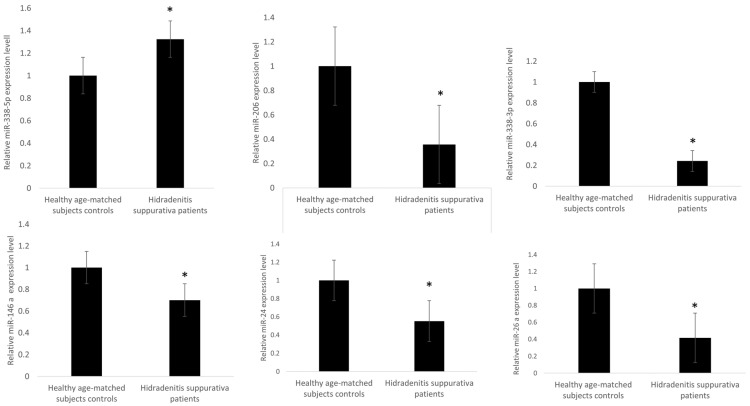
MicroRNA expression levels in blood leukocytes from HS patients versus healthy age-matched subjects. The expression of microRNAs was studied in blood leukocytes of HS patients, by microRNA assay-based quantitative real-time PCR following the delta–delta Ct method. Statistically significant differences were tested at * *p* < 0.05.

**Figure 2 genes-13-01544-f002:**
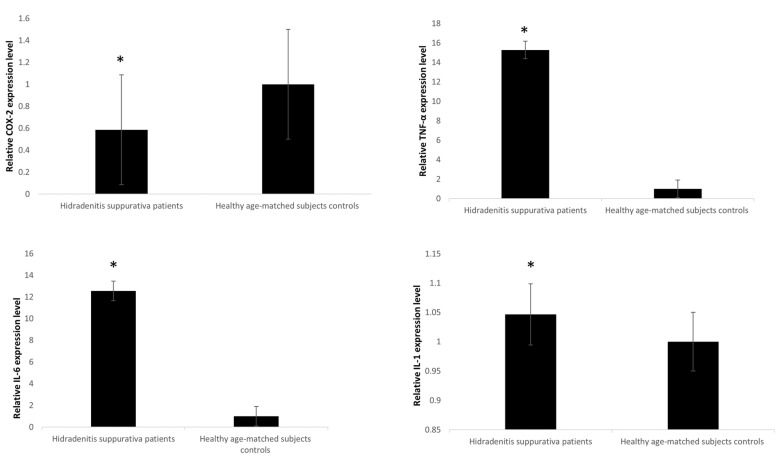
mRNA expression levels in blood leukocytes of HS patients versus healthy age-matched subjects. The expression of mRNAs was studied in blood leukocytes of HS patients by assay-based quantitative real-time PCR following the delta-delta Ct method. Statistically significant differences were tested at * *p* < 0.05.

**Figure 3 genes-13-01544-f003:**
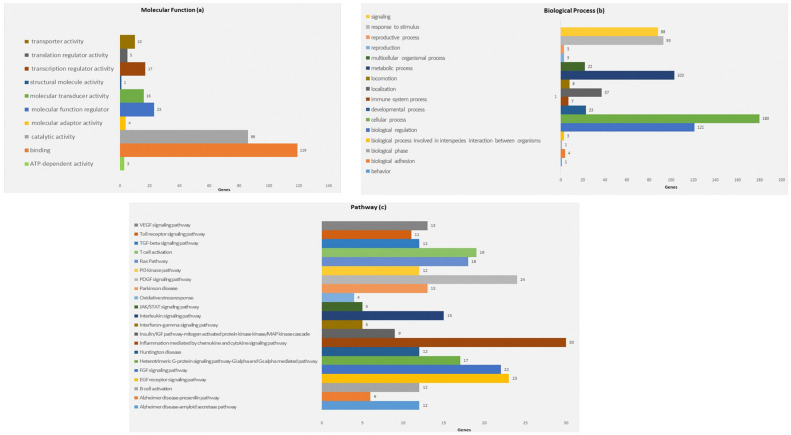
Panther gene ontology (GO) term-enrichment analysis for microRNA-associated genes in HS. Distribution of genes according to molecular function (**a**). Distribution of genes according to biological function (**b**). Distribution of genes according to the analysis of pathway (**c**). Beside each category, the percentage of gene frequency is reported. The number of assigned genes may be greater than the number of recognized genes as the same gene can be included in different categories.

**Figure 4 genes-13-01544-f004:**
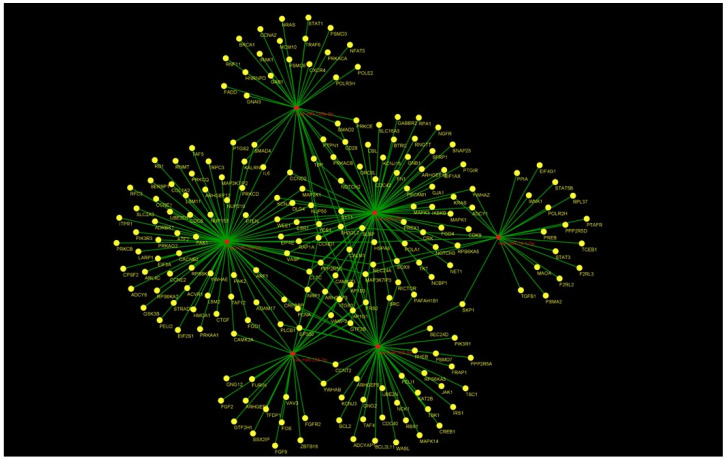
Combined molecular analysis in HS. Functional annotations of target genes, together with their miRNAs, are visualized as a network workflow (Cytoscape 3.6.0).

**Table 1 genes-13-01544-t001:** Demographic and clinical characteristics of HS patients. Data are presented as n (%) or mean ± SD.

	HS Patients (n. 25)	Controls (n. 12)	*p*-Value
**Sex n. F/M (%)**	19/6 (76)	5/7 (42)	
**Age mean (SD)**	25.71 ± 13.31	33.25 ± 11.26	0.06
**BMI mean (SD)**	28.11± 6.19	25.72 ± 2.98	0.23
**Smokers’ status n. (%)**	51%	-	
**Cholesterol (mg/dL)**	205.98 ± 38.57	155.72 ± 35.29	*0.003*
**Triglycerides (mg/dL)**	90.97 ± 27.16	85.63 ± 38.35	0.69
**Glycemia (mg/dL)**	96.57 ± 22.53	84.42 ± 5.38	0.18
**C-Reactive Protein (mg/L)**	10.57 ± 12.52	-	
**Hurley n. (%)**	15 (60)	-	
**Hurley II n. (%)**	8 (32)	-	
**Hurley III n. (%)**	2 (8)	-	

Anthropometric and biochemical parameters of study participants.

**Table 2 genes-13-01544-t002:** Panther Pathway of microRNA interacting genes.

Panther Pathway	*p*-Value	Molecules	miRNAs
Alzheimer’s disease–amyloid secretase pathway	4.40 × 10^11^	MAPK1, PRKACA, FURIN, ADAM17, PRKCE, PRKCQ, CACNB2, MAPK3, PRKCD, MAPK14, PAK1, PRKCB	hsa-miR-206, hsa-miR-146a-5p, hsa-miR-338-3p, hsa-miR-26a-5p, hsa-miR-338-5p
Alzheimer’s disease–presenilin pathway	3.55 × 10^3^	FURIN, NOTCH2, ADAM17, NOTCH3, TRPC3, GSK3B	hsa-miR-338-3p, hsa-miR-206, hsa-miR-26a-5p
Axon guidance mediated by Slit/Robo	3.22 × 10^3^	CXCR4, NET1, CDC42	hsa-miR-146a-5p, hsa-miR-206,
Axon guidance mediated by netrin	6.41 × 10^5^	NET1, PIK3R1, PIK3R3, VASP, CDC42,	hsa-miR-206, hsa-miR-338-5p, hsa-miR-26a-5p,
Axon guidance mediated by semaphorins	2.47 × 10^2^	NRP1, PAK1	hsa-miR-26a-5p, hsa-miR-338-3p, hsa-miR-338-5p, hsa-miR-206
FAS signaling pathway	3.13 × 10	FADD	hsa-miR-146a-5p
Inflammation mediated by chemokine and cytokine signaling pathway	5.72 × 10^21^	PREX1, CAMK2G, CXCR4, ITPR1, CAMK2A, MAPK1, PTEN, NFAT5, PAK2, PRKCE, PRKACB, PTGS2, MAPK3, GNG2, IKBKB, PTAFR, ADRBK1, STAT3, GNG12, CDC42, PLCB1, PRKACA, STAT1, GNAI3, KRAS, NRAS, IL6, PAK1, ADCY6, PRKCB,	hsa-miR-206, hsa-miR-338-3p, hsa-miR-146a-5p, hsa-miR-26a-5p, hsa-miR-338-5p, hsa-miR-24-1-5p
Insulin/IGF pathway-mitogen activated protein kinase kinase/MAP kinase cascade	5.00 × 10	MAPK1, RPS6KA3, MAPK3, PTGIR, RPS6KA6, RPS6KA5, IRS1, RPS6KA2, FOS	hsa-miR-338-5p, hsa-miR-206, hsa-miR-26a-5p, hsa-miR-338-3p
Insulin/IGF pathway-protein kinase B signaling cascade	9.03 × 10^6^	TSC1, PTEN, PIK3R1, PIK3R3, IRS1, GSK3B	hsa-miR-338-5p, hsa-miR-26a-5p
Interferon-γ signaling pathway	3.30 × 10^5^	MAPK1, JAK1, MAPK3, MAPK14, STAT1	hsa-miR-206, hsa-miR-338-5p, hsa-miR-146a-5p
Interleukin signaling pathway	3.38 × 10^13^	MAPK1, RPS6KA3, MAPK3, RPS6KA6, FRAP1, IKBKB, STAT3, STAT1, STAT5B, IRS1, RPS6KA2, FOS, NRAS, IL6, GSK3B	hsa-miR-206, hsa-miR-338-5p, hsa-miR-26a-5p, hsa-miR-24-1-5p, hsa-miR-146a-5p, hsa-miR-338-3p
Oxidative stress response	3.85 × 10^3^	ATF2, MAPK14, STAT1, BCL2	hsa-miR-26a-5p, hsa-miR-338-5p, hsa-miR-146a-5p

## Data Availability

For reasons of privacy and confidentiality, the data from this study are available from the corresponding authors upon reasonable request.

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
