# Peer review of "Circulating microRNAs in Hidradenitis Suppurativa"

_genes, 2022, doi:10.3390/genes13091544_

Round 1

Reviewer 1 Report

This article concerning miRNA and correlation to HS is of great medical and social interest. The article's structure is solid and the results are clearly presented. A few observations:

1) English should be revised (spelling, terms' appropriateness) and a grammar check should be performed (punctuation, word order).

2) Were pregnancy tests performed?

3) Smoking habit and miRNA expression can be related. This correlation should be addressed.

4) In the discussion paragraph, great focus has been paid to -146a-5p and on a side -338-5p. More information about the other miRNAs would prove interesting and skin related morbidities.

5) A few "study limitations" lines would prove useful.

Author Response

This article concerning miRNA and correlation to HS is of great medical and social interest. The article's structure is solid and the results are clearly presented. A few observations:

1) English should be revised (spelling, terms' appropriateness) and a grammar check should be performed (punctuation, word order).

Answer: we revised English in the text and checked grammar.

2) Were pregnancy tests performed?

Answer: Yes, we did. The patients were not pregnant.

3) Smoking habit and miRNA expression can be related. This correlation should be addressed.

Answer: we thank the reviewer for the interesting question. Smoking cessation could be a crucial part of the management of HS, however, in Table1 is reported the smokers status (51%) in a total of 25 patients. To calculate a correlation a larger number of matched patients would be necessary. Certainly, we aim to assess this correlation in the future.

4) In the discussion paragraph, great focus has been paid to -146a-5p and on a side -338-5p. More information about the other miRNAs would prove interesting and skin related morbidities.

Answer: we paid most attention on miR-146a-5p and miR-338-5p, because these microRNAs are particularly interesting and significant as an innate and adaptive immunity cell function and modulator of differentiation and inflammation. Indeed, as we wrote in the manuscript, miR-338-5p was closely related to HS invasiveness, production of cytokines, and atypically overexpressed. MiR-338-5p, may represent a good promise as non-invasive clinical biomarker for HS.

5) A few "study limitations" lines would prove useful.

Answer: we thank the reviewer for the interesting analysis. In agree with this, in the conclusion we highlighted the need for additional studies. We wrote:

Additional experiments could be performed to understand the molecular network in Hidradenitis and if other types of cells may contribute in development and progression of the disease. miR-338-5p has the prospective to be employed for the management and treatment of HS. Since miRNAs are found in body fluids, circulating miRNAs, as miR-338-5p, represent a good potential as clinical non-invasive biomarkers, even in HS. Supplementary research is needful to settle such assumption.

Reviewer 2 Report

The authors of the paper set out to investigate circulating microRNAs in HS.

Overall this is an interesting article.

It is noteworthy that Alzheimer pathway is enriched in panther pathway …. Could this be linked to GSC variation in recruited HS cohort? Was the cohort inclusive of patients with syndromic forms of HS? (recent https://doi.org/10.1111/jdv.18473 workhave excluded patients with syndromic HS). Would you care to comment on this?

Line 136 : “isolate 7 mature miRNA species” however figure 1 shows 6; (338-5p, 26a, 206. 338-3p, 24, 146a)

Figure 2 (legend, lines 178-180), refers to mRNA expression but shows COX2, TNAfa, Il-6, Il1. are these mRNA levels of the cytokines. Please clarigy.

The text in figure 3 is very difficult to read.

Line 203: Pace et PMID: 35401657  can be used as reference.

Line 214: “for the first time” … hessam et (2017) PMID: 28028756 have already alluded to 146a. It may serve as a reference.

Line 248 (combined with 204): Epigenetic investigations have been identified as potential blood based biomarkers for identification of disease having univariant correlation (PMID: 35044423) and may serve as a reference.

Author Response

The authors of the paper set out to investigate circulating microRNAs in HS.

Overall this is an interesting article.

1) It is noteworthy that Alzheimer pathway is enriched in panther pathway …. Could this be linked to GSC variation in recruited HS cohort? Was the cohort inclusive of patients with syndromic forms of HS? (recent https://doi.org/10.1111/jdv.18473 work have excluded patients with syndromic HS). Would you care to comment on this?

Answer: we thank the reviewer for the interesting question. We excluded patients with syndromic HS from the recruited HS cohort.

In the Materials and Methods we wrote: “Subjects were excluded from our study if they met any of the following criteria: age < 18 years, BMI<17 or >35, major metabolic disorders (Type 2 Diabetes, cardiovascular disorders, and metabolic syndrome), syndromic HS, the concomitant presence of an inflammatory cutaneous or systemic syndromes and the presence of cancer; were excluded also the patients receiving any systemic treatment which could interfere with the considered parameters. Disease staging was based on the three-degree Hurley’s scale.”

2) Line 136 : “isolate 7 mature miRNA species” however figure 1 shows 6; (338-5p, 26a, 206. 338-3p, 24, 146a)

Answer: we thank the reviewer for the question. Figure 1 shows all the microRNAs except let-7, because as reported in the Results: In all samples, expression level of let-7 was low and there was no significant differences between HS and healthy controls.

3) Figure 2 (legend, lines 178-180), refers to mRNA expression but shows COX2, TNAfa, Il-6, Il1. Are these mRNA levels of the cytokines. Please clarify.

Answer: Figure 2 refers to mRNA expression of the cytokines. The expression of mRNAs was studied in blood leukocytes of HS patients, by assay-based quantitative real-time PCR following the delta-delta Ct method. Statistically significant differences were tested at *p<0.05.

4) The text in figure 3 is very difficult to read.

Answer: The Figure 3 was a draft; we are to going to include the final figure to publish with higher resolution.

5) Line 203: Pace et PMID: 35401657  can be used as reference.

Answer: In agree with this, we used as reference : Pace NP, Mintoff D, Borg I. The Genomic Architecture of Hidradenitis Suppurativa-A Systematic Review. Front Genet. 2022 Mar 23;13:861241. doi: 10.3389/fgene.2022.861241. PMID: 35401657; PMCID: PMC8986338.

6) Line 214: “for the first time” … Hessam et (2017) PMID: 28028756 have already alluded to 146a. It may serve as a reference.

Answer: In agree with this, we cited Hessam et al (2017) as reference 23.

7) Line 248 (combined with 204): Epigenetic investigations have been identified as potential blood based biomarkers for identification of disease having univariant correlation (PMID: 35044423) and may serve as a reference.

Answer: In agree with this, we inserted as reference 22 the suggested reference (Der Sarkissian S, Hessam S, Kirby JS, Lowes MA, Mintoff D, Naik HB, Ring HC, Chandran NS, Frew JW. Identification of Biomarkers and Critical Evaluation of Biomarker Validation in Hidradenitis Suppurativa: A Systematic Review. JAMA Dermatol. 2022 Mar 1;158(3):300-313. doi: 10.1001/jamadermatol.2021.4926. Erratum in: JAMA Dermatol. 2022 May 1;158(5):590. PMID: 35044423; PMCID: PMC9131897.)